# Development and Validation of the Nursing Care and Religious Diversity Scale (NCRDS)

**DOI:** 10.3390/healthcare11131821

**Published:** 2023-06-21

**Authors:** Carla Murgia, Alessandro Stievano, Gennaro Rocco, Ippolito Notarnicola

**Affiliations:** 1Department of Biomedicine and Prevention, University of Rome “Tor Vergata”, 00133 Rome, Italy; carlamur@tiscali.it; 2Centre of Excellence for Nursing Scholarship, OPI, 00136 Rome, Italy; alessandro.stievano@gmail.com (A.S.); genna.rocco@gmail.com (G.R.); 3Department of Clinical and Experimental Medicine, University of Messina, 98100 Messina, Italy

**Keywords:** Confirmatory factor analysis, exploratory factor analysis, linguistic validation, nursing care, religious diversity, spirituality

## Abstract

(1) Background: In response to the impact of religious intervention on health outcomes and the importance of documenting how nurses experience the spiritual need of 392 hospitalized patients, it is vital to provide the nursing profession with instruments to evaluate these spiritual aspects. This study describes the development and validation of the Nursing Care and Religious Diversity Scale (NCRDS); (2) Methods: A two-step design was used for NCRDS translation and psychometric validation. The tool design was developed in the first step, while the psychometric characteristics were tested in the second step. An inductive study was conducted to test the validity and reliability of the NCRDS tool. The overall sample consisted of 317 nurses; (3) Results: The final instrument comprised 25 items in five dimensions. The construct validity indicated five dimensions. The face and content validity were adequate. Test-retest reliability displayed good stability, and internal consistency (Cronbach’s α) was acceptable (0.83); (4) Conclusions: Initial testing of the NCRDS suggested that it is a valid and reliable instrument to evaluate individuals in religious diversity, with five dimensions for evaluating the meaning of spirituality and individual belief, the religious healthcare environment, educational adequacy, spiritual and religious needs, and religious plurality.

## 1. Introduction

Some scholars point out that the human need for sacredness and religion in an era characterized by risk and uncertainty [1] makes the birth of new religious movements or the affirmation of other religions in countries evident. In the clinical-welfare field, a study conducted in the European Intensive Care Units highlighted how religious sentiment was considered an essential aspect of the human dimension of the person [2].

Several works describe the impact of religious intervention on health outcomes. A certain number of patients and healthcare professionals believe in the therapeutic power of prayer. Prayer can be successful in the recovery of patients in the hospital [3], in the acceptance of mandatory vaccinations [4], improves immune function [5], improves the course of rheumatoid arthritis and reduces anxiety [6], can alleviate suffering and promote good health [7], can alleviate the suffering of children and their families [8] can offer together with other religious rituals some hope when medical intervention no longer has a therapeutic effect [9].

In more than a few European countries, there is a large body of evidence regarding competence and formation in spiritual care [10,11,12,13]. The EPICC (Enhancing Nurses’ and Midwives’ Competence in Provide Spiritual Care through Innovative Education and Compassionate Care) project [14,15] has produced a standard of good practices for spiritual care education, providing a set of skills (knowledge, skills, and attitudes). However, the network of this international project has found a significant inconsistency in how spirituality is addressed within nursing and midwifery training programs across Europe. These authors recognized crucial limitations. All knowledge about spiritual care is not transferable or universally acceptable due to cultural and contextual differences among countries.

In the nursing degree in Italy, the student deepens different disciplinary areas, such as anthropology, sociology, and general pedagogy, social science disciplines, with contents almost unknown to the student population of the previous vocational schools. The study of these disciplines has undoubtedly involved a greater competence and awareness in the management of the different areas of nursing, allowing the new nurses to expand their weltanschauung for a more dynamic understanding of the world in which we live and to respond to the socio-cultural stimuli of a work environment destined to change in an increasingly multiethnic arena.

The anthropological formation, the rediscovery of interethnic relationships, and the overcoming of one’s ethnocentrism allow a better approach with users of other cultures and greater attention to spiritual and religious assistance [2,16,17].

In Italy in 2021, for the IDOS Immigration Statistical Dossier, the number of foreign residents was 5,013,200, with an incidence equal to 8.5% of the Italian population (an incidence higher than the average of the European Union, equal to 8.2%).

This is a crucial socio-religious assessment in the Italian context, where interest in formal religion decreases as in other European countries and Westernized nations (Canada, USA, Australia). Therefore, the complexity of the current Italian social context leads nurses to deal daily with an increasing variety of communities in terms of culture, language, and religion [18]. Considering the versatility of religions, this analysis can capture some of the “blurred” and “different” needs of people of all faiths and spirituality. It can make us understand the way spirituality is defined and why it is crucial and necessary nursing care in a multireligious society. Globally, nurses report that they need to have adequate preparation and training for spiritual care in their university programs. There is a need for scientific evidence to build a more coherent approach to how spiritual care skills can be achieved in education and nursing [19]. Therefore, in response to these concerns, given the importance of documenting how nurses experience the spiritual need of hospitalized patients who profess a religious belief different from Christian-Catholic or Christian-Protestant, this study describes the development and validation of a new instrument, “Religious Diversity and Nursing Care”. We propose using this new tool to offer an opportunity for nurses to broaden their understanding and skills on the human condition and dignity so that they acquire greater awareness of different religions and can also regard people with different religious affiliations differently. The development of this instrument has been an integral part of a research project in some Roman hospitals and could represent one moving forward step for assessing the spiritual need of patients of unfamiliar cultures and religions. Furthermore, this tool could indicate how spiritual assistance influences care processes, particularly how healthcare professionals take care in other healthcare contexts of international mobility [20,21]. This empirical study has aimed to develop an inventory to evaluate nursing care and religious diversity in an Italian nursing professional context and assess its psychometric properties.

## 2. Materials and Methods

### 2.1. Study Design

This study was divided into two phases: in phase one, item development was set up, while in the second phase, the psychometric characteristics of the instrument were tested.

### 2.2. Phase 1: Step 1

The first step of Phase 1 consisted in defining spirituality and religious diversity to understand the global spirituality of nurses in coping with a range of challenging work situations that they typically face in practice. Two focus groups were conducted between May and July 2021 with nurses from the Rome area, involving ten nurses per session [22]. Participants in this study were enrolled using available sampling to include nurses from different specialties and departments with more than five years of experience in the same occupational field. All participants were asked to provide their written consent to be involved in the focus group discussions and to be audio recorded.

For each focus group, a primary question was elaborated to determine which nurses’ circumstances were most challenging in their daily clinical practice to create a basis for developing a valuable tool for measuring the spirituality and religious diversity of the nursing profession.

Each focus group was led by an experienced facilitator who began each session with a question guide to stimulate discussion. Each nurse participating in the focus groups could leave the discussion anytime, and confidentiality was guaranteed. In addition, each session lasted between 1 and 2 h, including the initial readings; the transcription of the recordings took place verbatim. To ensure each participant’s anonymity, the transcriber used random numbers generated by special software on a website (random.org (accessed on 16 May 2023)) rather than entering the names of the nurses participating in the study.

A content analysis by Vaismoradi et al. [23,24] was performed. Focus group transcripts, to identify categories and key themes, were conceptualized into items to be included in the tool.

The authors used Nvivo software version 10 (QSR International, Denver, CO, USA) for data organization, providing coding and categorization as data reduction methods. The main themes were identified and codified by selecting critical statements from the transcriptions. The transcripts were reviewed by each study author individually; peer debriefing and member audits were performed to establish the reliability of the content [25,26].

Two main themes were identified: ‘spirituality in nursing quality’ and ‘religious diversity in nursing’. The results of the focus groups, including the quotations, were translated from Italian into English, following the recommendations to obtain a preferable reproduction more in line with the interpreted experiential experience of the participants [27].

### 2.3. Phase 1: Step 2

The development of the items of the scale of nursing care and religious diversity constituted the second step of phase 1; in fact, using the results produced by the focus groups, a further review of the literature was set up on international databases (PubMed, Cinahl, and Scopus), also assisted by empirical research carried out both in libraries and on social networks in the nursing sector. The database search included studies in English and Italian relating to the context analyzed. Remarkably, the literature review provided more construct on the focus group findings as we found a convergence of meanings. Thus, we developed an initial set of topics and revised them several times during consensus discussions among the authors.

The initial scale was composed of 33 items rated on a 5-point Likert scale (from “1-never” to “5-very often”). The total score ranges from 0 to 100, with higher scores representing greater religious plurality and spirituality. After revisions, some unclear and repetitive items were removed after being evaluated by nursing experts consulted to assess face validity. Consequently, the final initial scale was composed of 33 items grouped into three subscales, Section 1: the health organization of the operating unit; Section 2: spiritual assistance towards the other; Section 3: spiritual and religious expression.

### 2.4. Phase 2: Psychometric Testing

To ensure that the developed tool was suitable for measuring nurses’ spirituality and religious diversity, we tested the validity and reliability of the Nursing Care and Religious Diversity Scale.

#### 2.4.1. Participants and Setting

A convenience sampling method was used. This study was conducted in two major hospitals in Rome, Italy. The final sample comprised 317 nurses, and data were collected between June 2022 and December 2022 (430 nurses were invited, and 73.72% filled out the questionnaires).

#### 2.4.2. Data Analysis

As an introductory analysis, descriptive statistics were compiled for the demographic characteristics of the analyzed sample, including skewness and kurtosis indices, to ascertain normality. The Nursing Care and Religious Diversity Scale (NCRDS) was tested for Face and Content Validity (CV) in the form of a Content Validity Index (CVI) [28,29].

Construct validity was assessed by Exploratory Factor Analysis (EFA) and Confirmatory Factor Analysis (CFA). To verify the normality of the data, the Kolmogorov–Smirnov test was calculated for the variables examined.

EFA was performed using the maximum likelihood method in which eigenvalue analysis and scree test were used to select the number of factors to extract. Items whose loading value was more significant than 0.30 were kept. Before proceeding with the EFA, the Bartlett test and the Kaiser-Meyer-Olkin index were examined to evaluate the factorizability of the correlation matrix. Cronbach’s α coefficient was used to assess internal consistency. The following fit indices were considered to evaluate the CFA model: omnibus fit indices such as chi-square (χ^2^), incremental fit indices such as CFI (values > 0.95 indicated a good fit), RMSEA (values < 0.06 indicated a good fit), and weighted root mean square residue (WRMR; values 1.0 indicated a good fit). To determine stability, reliability was measured using the test-retest method [30]. All statistics were calculated using SPSS 22 software (SPSS, Inc., Chicago, IL, USA) and Mplus 7.1 software (Muthén and Muthén, 1998–2012) (Appendix A).

### 2.5. Ethical Considerations

This study was ethically approved by the Center of Excellence for Nursing Scholarship OPI Rome protocol number 2.21.22 in accordance with international ethical principles and Italian legal and research ethics requirements for non-interventional studies.

Participants were informed that participation was voluntary and that they could withdraw or refuse to participate at any time without any consequences from the directors of the hospitals. Participants were also informed about the confidentiality of their responses and anonymity in data elaboration for the study’s final report. This study did not involve patients. The study was designed, conducted, recorded, and reported on consistently with the international ethical and scientific quality standards indicated by good clinical practice (GCP) and standard operating procedures (SOPs). Participants were asked to provide written informed consent.

## 3. Results

### 3.1. Sample Characteristics

Our sample consisted of nurses (*n* = 317). Most nurses were female (69.1%). 21.5% were between the ages of 40 and 44. 26.5% had several years working as a nurse ranging from 16 to 20 years. 94% of the sample analyzed was Italian, and 47% were married. The clinical areas in which nurses operated were as follows: medical area = 39.1%; surgical area = 33.8%; the remaining sample, 27.1%, did not specify the clinical area to which they belonged (Table 1).

### 3.2. Explore the Effects of Spirituality and Religious Beliefs in Clinical Settings

We used a one-way ANOVA test to determine the effect of contextual variables in the clinical areas of the analyzed sample. The results (see Table 2) indicated that the type of clinical environment had good levels on the spirituality and religious beliefs of the nurses, particularly in the surgical area (F1 = 64.299; F2 = 68.474; F4 = 64.813; F5 = 61.628), where only four factors on the NCRDS outperformed the Medicine area; which instead obtained a lower score (F1 = 58.871; F2 = 62.231; F4 = 60.363; F5 = 53.613). Factor three instead resulted in almost the same in both clinical areas (CF3 = 44.065 and MF3 = 44.355). For one-way ANOVA, a *p*-value < 0.05 was set for statistical significance.

### 3.3. Item Descriptive Statistics

Table 3 shows the descriptive statistics of each Nursing Care and Religious Diversity Scale (NCRDS) element, including mean, standard deviation (DS), asymmetry, and kurtosis. What is more, the descriptive analysis showed that while there was a trend toward positive responses, item averages were often not too high. Furthermore, there was sufficient variance in the score, suggesting that participants were unlikely to respond in a socially desirable manner. Considering that the elements of the NCRDS have a 5-point response format on the Likert scale and that most items have asymmetry and kurtosis indices within |1|, they could be treated as continuous variables [31].

### 3.4. Validation

#### 3.4.1. Face and Content Validity

The initial scheme of the instrument with the 33 items identified in the second step of phase 1 and grouped into four subscales was sent to 13 nursing experts (recruited in universities and hospitals). Their Content Validity Index (CVI) evaluation was provided through a 4-point Likert scale (1 = not relevant; 2 = quite relevant; 3 = relevant; 4 = very relevant). Determining the CVI item and reporting an overall CVI are important components needed for tools, especially when the tool is used to measure health outcomes or guide clinical decision-making [32]. Content validity provides evidence about the degree to which elements of an assessment tool are relevant and representative of the construct being targeted for a particular assessment purpose [33]. The experts also had to answer some open-ended questions to investigate the clarity of the items analyzed and recommend any additional items to be included in the scale. The CVI was calculated to evaluate the items [31,32]. Eight items were eliminated, as their relevance was less than 0.75. showing redundancy with other items on the same scale. Based on the indications provided by the experts, the authors also carried out textual revisions of the items. The mean CVI was acceptable at 0.87 (SD = 0.13), excluding the eight redundant items. For redundant items, we meant items that had similar meanings and could be considered overlapping. Those elements were thought superfluous and thereby eliminated.

The 25-item final draft was tested for apparent validity by another group of six experienced nurses, and the authors revised the items based on their feedback, reformulating and eliminating dubious and repetitive words. The final version of the Nursing Care and Religious Diversity Inventory Scale consisted of 25 items which were grouped into three subscales.

#### 3.4.2. Construct Validity

We performed an EFA with the maximum likelihood method with Varimax rotation to examine the new scale’s dimensionality on the whole study sample (*n* = 317). Bartlett’s sphericity test was significant (χ^2^ = 4360.2. df = 406. *p* < 0.00), and the Kaiser–Meyer–Olkin test was 0.782. Some authors suggest that for identifying which elements are associated with which factors, both approaches are effective. However, to identify a simple structure when it is present, the oblique method is preferable. In our study, we used Varimax rotation because NCRDS has a complex structure [34]. These results showed that the correlation matrix was suitable for performing factor analysis. In line with our hypothesis, a parallel analysis was performed, which suggested the extraction of 5 dimensions [35,36]. These factors, after rotation, explained 20.6%, 12.2%, 10.5%, 7.3%, and 5.6% of the common variance, or 56.3% of the overall variance. The loading factors (Table 4) were all greater than 0.30. These results suggested that the model was the most suitable to cross-validate with CFA. The CFA was performed on the whole sample (n = 317) and the processed results confirmed the adequacy of the multidimensional model with a satisfactory fit to the data. χ^2^ (n = 317) = 1032.82. DF = 256; *p* < 0.00; CFI = 0.77; NFI = 0.72; RMSEA = 0.09 (90% CI = 0.092–0.104). *p* = 0.00; WRMR = 0.97 (Figure 1).

In the CFA of our study, the goal was to obtain the adaptation statistics to reach a predetermined level; therefore, we tried to add related errors until we obtained an “acceptable” adaptation of the model. This can be performed by considering that the related measurement error can be modeled in a CFA solution provided that this specification is justified and other identification requirements are met [37]. According to Brown and Moore, when specifying that the measurement error is random (that is, the unique variances of the indicator are not correlated), the hypothesis is that the observed relationship between any two indicators that load on the same factor is entirely due to the shared influence of the latent variable. Or rather, if the factors were partial, the correlation of indicators would be zero è [38]. However, in CFA, the specification of related errors may be justified based on the effects of the method reflecting the additional co-variation of the indicator resulting from common assessment methods, such as observer evaluations or questionnaires [39].

### 3.5. Reliability

Cronbach’s alpha related to internal consistency was evaluated after CFA to assess whether items in each subscale measured the same construct. As shown in Table 3, Cronbach’s alpha in the dimension Meaning of Spirituality and Individual Belief (6 items) was 0.82, and in the Religious Healthcare Environment (6 items) was 0.82. Educational Adequacy (4 items) was 0.83, Spiritual and Religious Needs (4 items) was 0.82, and Religious Plurality (5 items) was 0.83. All scale dimensions were deemed acceptable as exceeding the threshold of 0.80 [40]. Moreover, considering the whole instrument, it had a Cronbach’s alpha of 0.83.

Thanks to a nurse identifier, through a code, the authors were able to perform the test-retest. Several nurses (*n* = 30) were asked to complete the questionnaire again after two weeks had elapsed. The test-retest method was used to estimate the stability of the tool. The response rate for the test-retest was 100%. The Pearson correlation coefficient between the two measurements was 0.92 (*p* < 0.001), and the scores had a normal distribution. as shown by a Kolmogorov–Smirnov test.

## 4. Discussion

This study aimed to develop and validate an instrument to assess nursing care and religious diversity. The theoretical framework of this empirical study was based on previous work performed by the authors on spirituality and religious diversity [16]. According to Scholz et al. [41], the research question seeks to see how the setting where the scale was developed may have influenced the development of the tool and the results. Making a hypothesis on the above, if a researcher wants to know nurses’ perceptions of their spirituality, a limited set of items is needed, which will probably be little influenced by clinical practice. However, if a researcher is to study the general sense of nurses in the religious diversity of caregiving, a broader exploration is needed and will likely be more influenced by different clinical contexts.

For this reason, the instrument outlined in this study was developed starting from an Italian professional context, even if the items of this instrument were developed considering both the focus group results and a broader literature review. The Italian nursing context shares some similarities in spirituality and religious diversity with other European contexts. However, some peculiarities of the Italian environment make the tool developed specific and sensitive to the Italian setting.

Collectively, the elements developed were intended to explore some aspects of the complicated clinical contexts that might occur during a working day. For this reason, the focus group results facilitated the identification of three main themes: the health organization of the operating unit, spiritual assistance towards the other, and spiritual and religious expression.

Burns and Grove [42] stated that content validity was relevant to ensure congruence between the research scope and the developed instrument. Additionally, notable was the role of the experts with the judgment and feedback they gave in assessing both content and face validity, showing a high degree of agreement (mean = 0.89; SD = 0.14). Indeed, the assessment of apparent validity provided helpful revisions of the formulation for the final tool with 25 elements grouped into five dimensions based on the results of the focus groups and the analysis of the literature investigated in the different databases queried: meaning of spirituality and individual belief (6 items), religious healthcare environment (6 items), educational adequacy (4 items), spiritual and religious needs (4 items), and religious plurality (5 items).

Based on the acquired focus group results, EFA extracting five factors with a Varimax rotation was a valid model, explaining 56.3% of the extracted variance. Burns and Grove [42] suggest that the sample size for performing an EFA is five to ten participants per variable to ensure sufficient stability for psychometric testing. Our sample (n = 317) was much larger than this minimum requirement; thus, our EFA model was adequate for the analysis. Some commonality loading factors were not greater than 0.30, meaning there were items requiring elimination. Hence, eight items from the original 33 scale were deleted, and only 25 items scoring above 0.30 were considered valid. The CFA performed on the sample (*n* = 317) was used to cross-validate the most plausible factor structure model derived from the EFA and confirmed a multidimensional model with a satisfactory fit to the data, explaining 56.3% of the total variance.

In the first dimension, meaning of spirituality and individual belief (6 items), for example, the item describing if the environment where nurses are working had signs and symbols linked to the Catholic Christian religion, it was clearly stated that nurses manage best practices in providing spiritual assistance based on their individual principles and on the beliefs of patients. From the focus group results, it emerged that nurses had to overcome structural and organizational irresolution in their everyday clinical practice. This may also be due to moral issues that nurses face in different clinical contexts; therefore, these obstacles could create different moral and ethical choices in the nurses’ experience [43].

In the second dimension, religious healthcare environment (6 items), it was emphasized how the nurse had to possess the skills to offer patients care in full respect of their cultural, human, and spiritual principles, such as to be able to experience the state of illness, its evolution and the outcomes of health treatments within a nurse/patient relationship characterized by support [44]. For example, in the items it is explored whether the patient needs counseling, support, human and spiritual help of a specific type carried out by members of the patient’s cult and how spiritual needs can be linked to the religion he professes.

In the third dimension, educational adequacy (4 items), we investigated how training could be designed by structuring it organically in the three fundamental directions of an ideal systemic model, bearing in mind the multiple aspects of daily reality that the nurse had to face in clinical contexts [45,46]. In the items, it is explored whether the nurse is adequately trained in identifying the patient’s deep meanings in his state of illness/discomfort through an assessment of spiritual needs and if he has good skills to support and support the patient with respect to his spiritual needs regardless of the patient’s religion.

In the fourth dimension, spiritual and religious needs (4 items), we explored that vigilant and attentive spiritual care is increasingly recognized as a fundamental part of high-quality nursing care. For instance, these items that make up this factor explore the work of the nurse who should devote himself more to the patient’s spiritual needs since spiritual well-being is an important component of the person’s health; therefore, it is important to satisfy the spiritual and religious needs related to the state of disease, as these are secondary to the physical needs of the patient. Therefore, spirituality must be considered one of the patient’s vital signs and must be regularly evaluated [47]. More knowledge on how spiritual needs can change the progression of the disease emerged as a necessity [48].

Finally, in the fifth dimension, Religious Plurality (5 items), a crucial nursing concept was explored. In fact, it should be possible for nurses to overcome the theoretical differences among religions and the existing interpretative conflicts since these conflicts are often generated within the same religion [49]. This is a major finding that is sometimes overlooked in this type of scale. In fact, as it was found in [50,51], within the same religions and spiritual viewpoints, many interpretive skirmishes are frequently ignored because people tend to think that possible incomprehension can only be found among completely different faiths.

### Limitations

Some essential limitations in this study must be observed when attempting to make sense of the results. The strong influence of the Italian context is the main limitation of the nursing profession’s nursing care and religious diversity scale. This is a valid tool for Italian nurses and could be suitable internationally, but it needs to be tested with more inductive research in different clinical contexts to evaluate its validity also in other environments.

Furthermore, participants were enrolled using convenience sampling, and data were collected via a crosscutting approach. For this reason, caution is required when generalizing the results. Further, reliability was assessed only by evaluating the internal consistency of the domains, with no information on stability. For this reason, future research should examine the indices of stability of the NCRDS over time.

## 5. Conclusions

This psychometric study developed and validated a 25-item tool for measuring nursing care and religious diversity in nursing. The results highlighted internal consistency, reliability, content validity, and construct validity. It will be essential to test the validity of the NCRDS in a context other than the Italian one to be replicated in different languages and contexts. The developed tool could be used to assess nursing care and religious diversity in different clinical contexts. Using this tool, nurses could improve the attitude of religiosity towards the patients they care for, while nursing leaders could gain a detailed understanding of assisting in religious diversity care.

## Figures and Tables

**Figure 1 healthcare-11-01821-f001:**
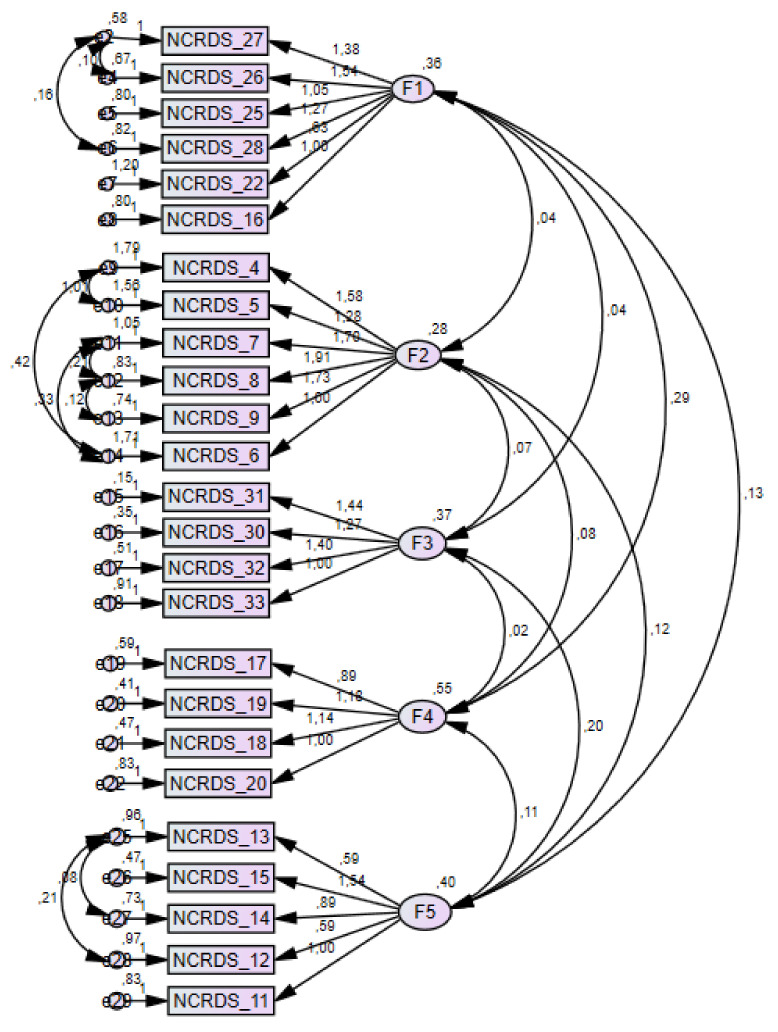
Structure of the NCRDS based on Confirmatory Factor Analysis.

**Table 1 healthcare-11-01821-t001:** Socio-demographic data of the sample.

Nationality	N	%
Brazilian	3	0.9
Italian	298	94.0
Romanian	2	0.6
No answer	10	3.2
Other	4	1.3
**Gender**		
Male	98	30.9
Female	219	69.1
**Age**		
Up to 24 years	6	1.9
25–29 years	12	3.8
30–34 years	28	8.8
35–39 years	55	17.4
40–44 years	68	21.5
45–49 years	64	20.2
50–54 years	50	15.8
55–59 years	32	10.1
60 or more years	2	0.6
**Civil status**		
Unmarried	77	24.3
Married	149	47.0
Cohabiting	48	15.1
Separated or divorced	41	12.9
Widow/widower	2	0.6
**Seniority of role**		
Up to 5 years	28	8.8
6–10 years	33	10.4
11–15 years	64	20.2
16–20 years	84	26.5
21–25 years	52	16.4
26–30 years	31	9.8
31 or more years	25	7.9
**In which operative unit do you work?**		
Medicine	124	39.1
Surgery	107	33.8
No answer	86	27.1
Total	317	100.0

**Table 2 healthcare-11-01821-t002:** ANOVA test in the clinical areas of the analyzed sample *.

		Mean	SD	SE	95% Confidence Interval for the Mean		Min	Max	F	*p*
					Lower Limit	Upper Limit				
Factor 1	Medicine	58.871	18.023	1.618	55.667	62.075	20.00	93.33	7.202	0.001
	Surgery	64.299	14.187	1.371	61.580	67.018	23.33	100.00		
	No answer	56.124	12.565	1.355	53.430	58.818	26.67	100.00		
	Total	59.958	15.724	0.883	58.220	61.696	20.00	100.00		
Factor 2	Medicine	62.231	20.468	1.838	58.593	65.869	20.00	100.00	8.700	0.000
	Surgery	68.474	19.470	1.882	64.742	72.205	30.00	100.00		
	No answer	56.744	18.171	1.959	52.848	60.640	26.67	93.33		
	Total	62.850	20.001	1.123	60.639	65.060	20.00	100.00		
Factor 3	Medicine	44.355	18.741	1.683	41.024	47.686	20.00	90.00	4.158	0.017
	Surgery	44.065	18.685	1.806	40.484	47.647	20.00	100.00		
	No answer	38.081	10.380	1.119	35.856	40.307	20.00	60.00		
	Total	42.555	17.038	0.957	40.672	44.438	20.00	100.00		
Factor 4	Medicine	60.363	18.980	1.704	56.989	63.737	20.00	100.00	10.521	0.000
	Surgery	64.813	15.795	1.527	61.786	67.840	25.00	100.00		
	No answer	53.721	14.167	1.528	50.684	56.758	20.00	95.00		
	Total	60.063	17.220	0.967	58.160	61.966	20.00	100.00		
Factor 5	Medicine	53.613	15.293	1.373	50.894	56.331	20.00	92.00	8.192	0.000
	Surgery	57.047	13.541	1.309	54.451	59.642	28.00	100.00		
	No answer	61.628	12.992	1.401	58.842	64.413	20.00	100.00		
	Total	56.946	14.431	0.811	55.352	58.541	20.00	100.00		

* Sample data analyzed in clinical areas: Medicine = 124; Surgery = 107; No answer = 86.

**Table 3 healthcare-11-01821-t003:** Item descriptive statistics.

	Mean	SD	Skewness	Kurtosis
NCRDS 1	3.15	1.129	−0.049	−0.511
NCRDS 2	3.04	1.237	0.059	−0.904
NCRDS 3	2.92	1.095	0.172	−0.515
NCRDS 4	3.10	1.186	−0.082	−0.713
NCRDS 5	2.72	1.161	0.148	−0.656
NCRDS 6	3.06	1.081	0.190	−0.451
NCRDS 7	3.28	1.603	−0.197	−1.576
NCRDS 8	3.78	1.419	−0.754	−0.845
NCRDS 9	2.98	1.361	0.097	−1.175
NCRDIS10	2.68	1.360	0.320	−1.065
NCRDIS11	2.29	1.252	0.712	−0.524
NCRDIS12	3.83	1.405	−0.822	−0.756
NCRDIS13	2.17	0.965	0.574	−0.032
NCRDIS14	2.24	0.974	0.496	−0.006
NCRDIS15	2.27	1.118	0.645	−0.257
NCRDIS16	1.83	1.135	1.137	0.179
NCRDIS17	2.79	1.013	0.262	−0.014
NCRDIS18	3.12	1.081	0.107	−0.486
NCRDIS19	2.94	1.074	0.126	−0.333
NCRDIS20	3.16	1.174	−0.027	−0.688
NCRDIS21	2.74	1.046	0.230	−0.484
NCRDIS22	2.84	1.192	−0.013	−0.786
NCRDIS23	2.65	1.022	0.414	−0.028
NCRDIS24	3.25	1.054	−0.130	−0.359
NCRDIS25	2.76	1.110	0.171	−0.561

**Table 4 healthcare-11-01821-t004:** Factor loadings for five extracted factors after Varimax rotation (*n* = 317).

	Item	F1	F2	F3	F4	F5
Factor 1 (6 item)	NCRDS_27	0.844	−0.062	−0.041	0.070	−0.100
	NCRDS_26	0.721	−0.103	−0.116	0.150	−0.045
	NCRDS_25	0.601	0.127	0.201	−0.012	0.114
	NCRDS_28	0.594	0.094	−0.001	0.124	0.000
	NCRDS_22	0.441	−0.261	0.018	0.012	0.096
	NCRDS_16	0.345	0.012	−0.188	0.307	0.097
Factor 2 (6 item)	NCRDS_4	0.089	0.776	−0.229	−0.086	0.196
	NCRDS_5	0.055	0.770	−0.180	−0.157	0.132
	NCRDS_7	−0.075	0.729	0.182	0.059	−0.139
	NCRDS_8	−0.106	0.711	0.170	0.002	0.105
	NCRDS_9	−0.185	0.676	0.056	0.231	0.040
	NCRDS_6	0.254	0.654	−0.244	−0.021	−0.212
Factor 3 (4 item)	NCRDS_31	0.068	−0.052	0.875	0.018	0.020
	NCRDS_30	0.080	−0.011	0.830	−0.061	−0.002
	NCRDS_32	0.020	−0.119	0.803	−0.042	0.072
	NCRDS_33	0.063	0.256	0.642	0.148	−0.087
Factor 4 (4 item)	NCRDS_17	−0.201	0.111	0.178	0.838	0.046
	NCRDS_19	0.053	−0.011	−0.031	0.813	0.031
	NCRDS_18	0.139	−0.066	−0.054	0.791	−0.111
	NCRDS_20	0.172	0.019	−0.159	0.612	0.062
Factor 5 (5 item)	NCRDS_13	−0.178	0.009	−0.054	−0.041	0.809
	NCRDS_15	0.071	0.106	0.294	−0.016	0.615
	NCRDS_14	0.004	−0.034	−0.068	0.183	0.581
	NCRDS_12	0.183	0.177	0.008	−0.108	0.515
	NCRDS_11	0.053	0.005	0.078	0.086	0.498

## Data Availability

Please email ippo66@live.com if you have any inquiries about data details supporting the reported results, including links to publicly archived datasets analyzed or generated during our study.

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
