# Peer review of "Development and Validation of the Nursing Care and Religious Diversity Scale (NCRDS)"

_healthcare, 2023, doi:10.3390/healthcare11131821_

Round 1

Reviewer 1 Report

The development of the scale is well designed and reported in detail. The performance of the scale has been tested in the most important ways and suggests a high standard of trustworthiness. 

It would be useful to have at least one example of an item contributing to each of the five dimensions, as the general description of the features of each item is quite brief. 

There are several scales covering spirituality and nursing care. The original contribution of this scale is in its inclusion of the Religious Diversity. The conclusion of the discussion (line 314) that "it should be possible for nurses to overcome the theoretical differences among religions and the existing interpretative conflicts since these conflicts are often generated within the same religion" is a major finding that needs proper backing and explanation.  I would expect therefore to see quite a lot more detail about this dimension, which is typically overlooked in scales of this type. 

Author Response

REPLIES TO REVIEWER 1

The development of the scale is well designed and reported in detail. The performance of the scale has been tested in the most important ways and suggests a high standard of trustworthiness. 

  1. It would be useful to have at least one example of an item contributing to each of the five dimensions, as the general description of the features of each item is quite brief. 

A: Based on the reviewer's instructions, we have included a brief description of an item for each dimension.

  1. There are several scales covering spirituality and nursing care. The original contribution of this scale is in its inclusion of the Religious Diversity. The conclusion of the discussion (line 314) that "it should be possible for nurses to overcome the theoretical differences among religions and the existing interpretative conflicts since these conflicts are often generated within the same religion" is a major finding that needs proper backing and explanation.  I would expect therefore to see quite a lot more detail about this dimension, which is typically overlooked in scales of this type. 

A: Based on the reviewer's advice, we have improved the paragraph by providing the reader with a better interpretation of this dimension in our study.

Reviewer 2 Report

The submitted manuscript describes the development and analysis of the nursing care and religious diversity scale (NCRDS) questionnaire. The paper is well-structured, and the analysis sounds convincing.
Major comments:
1.    In the introduction, I felt that too often, “some” is used when characterizing substantives. For example, I found “some scholars” (28), “some works” (33), “some patients” (33), “some countries” (41), etc. Please use more concrete writing or substitute “some” with another word in some cases.
2.    91: “we developed the instrument’s structure”. Would it not be more precise to speak about “item writing” or “item development”? In the same manner, the phrase “item production” (128) sounds unusual.
3.    137: It was unclear to me what is meant by a Likert format on a scale from 0 to 100. In the next sentence, it was said that answer options ranged from “never” to “always”. Please clarify.
4.    Sect. 3.1.: Include descriptive statistics about age (i.e., mean, standard deviation, and range).
5.    Table 1: Use the same number of digits after the decimal.
6.    Table 1: Write “skewness” instead of “asymmetry”.
7.    Table 1: The columns “min” and “max” can be excluded because they do not contain additional information. Include frequencies of all five categories in the table.
8.    Throughout the manuscript: Unfortunately, the authors switch between “old item names” and “new item names”. In order to ease reading, please always use the original item names (also used in Figure 1).
9.    198: Note that one can always justify using CFA with a normal distribution assumption, although the values of the items are discrete ranged between 1 and 5 (see https://doi.org/10.3389/feduc.2020.589965).
10.    Sect. 3: Please provide the covariance matrix of all 33 original items in an appendix, a supplement, or an OSF repository. With this information, researchers were able to reproduce your analysis.
11.    Sect. 3.3.1: It was said that seven “redundant items” were removed. First, what is meant by “redundant”? Second, I was confused because removing 7 items from the original test consisting of 33 items would not result in 25 items.
12.    207: Please provide more details about the computation of the CVI index because this is not a standard psychometric procedure. It might also be useful to move such a description into an appendix.
13.    Sect. 3.3.2: I would always prefer an oblique rotation method like promax over an orthogonal rotation method like varimax.
14.    241: When reporting a reliability measure of the whole scale, it might be more viable to use stratified alpha or omega total based on a bifactor model as reliability measures.
15.    247: The application of the Kolmogorov-Smirnov test seems questionable because it is unclear which procedure a normal distribution would be required.
Minor comments:
16.    Please provide references to used software packages (117, 169).
17.    184: Remove “(n=317)” from the title and include it in the text.
18.    194: write “What is more”.
19.    237: write “Cronbach’s alpha”
20.    270: write “Burns and Grove”
21.    285: write “were not greater”

Author Response

REPLIES TO REVIEWER 2

The submitted manuscript describes the development and analysis of the nursing care and religious diversity scale (NCRDS) questionnaire. The paper is well-structured, and the analysis sounds convincing.
Major comments:
1.    In the introduction, I felt that too often, “some” is used when characterizing substantives. For example, I found “some scholars” (28), “some works” (33), “some patients” (33), “some countries” (41), etc. Please use more concrete writing or substitute “some” with another word in some cases.

A: Based on the reviewer's observations, we have modified the introduction by inserting some synonyms, making the speech more fluid.

  1. 91: “we developed the instrument’s structure”. Would it not be more precise to speak about “item writing” or “item development”? In the same manner, the phrase “item production” (128) sounds unusual.

A: We have improved the paragraph based on the reviewer's advice.

  1. 137: It was unclear to me what is meant by a Likert format on a scale from 0 to 100. In the next sentence, it was said that answer options ranged from “never” to “always”. Please clarify.

A: We have rewritten the paragraph based on the reviewer's considerations, providing clarity to the interpretation of the likert scale and its score.

  1. Sect. 3.1.: Include descriptive statistics about age (i.e., mean, standard deviation, and range). TABLE 1

A: Based on the advice of the auditor we have included a new table, TABLE 1, with the demographic data of the sample.

  1. Table 1: Use the same number of digits after the decimal.

A: We have made the requested changes.

  1. Table 1: Write “skewness” instead of “asymmetry”.

A: We made the change based on the advice given by the reviewer.

  1. Table 1: The columns “min” and “max” can be excluded because they do not contain additional information. Include frequencies of all five categories in the table.

A: based on the reviewer's suggestion we excluded min and max, we also included in the appendix the frequencies of the five categories, providing the reader with a better understanding of the study

  1. Throughout the manuscript: Unfortunately, the authors switch between “old item names” and “new item names”. In order to ease reading, please always use the original item names (also used in Figure 1).

A: As recommended by the reviewer we have removed from Table 3 the words "old item names" and "new item names", in this way we have improved the comprehension for readers.

  1. 9.    198: Note that one can always justify using CFA with a normal distribution assumption, although the values of the items are discrete ranged between 1 and 5 (see https://doi.org/10.3389/feduc.2020.589965).

A: We have included bibliographic citation as recommended by the reviewer

  1. Sect. 3: Please provide the covariance matrix of all 33 original items in an appendix, a supplement, or an OSF repository. With this information, researchers were able to reproduce your analysis.

A: We have included the covariance matrix in the appendix as recommended by the reviewer

  1. Sect. 3.3.1: It was said that seven “redundant items” were removed. First, what is meant by “redundant”? Second, I was confused because removing 7 items from the original test consisting of 33 items would not result in 25 items.

A: As for the deleted items there was a typo: 8 those removed In this way, we got a scale of 25 items. Based on the reviewer's advice, we explained better in the paragraph what we meant by deleted redundant items.

  1. 207: Please provide more details about the computation of the CVI index because this is not a standard psychometric procedure.

R: As recommended by the reviewer we have inserted a new statement that better describe the CVI index, in order to give the readers a better explanation.

  1. Sect. 3.3.2: I would always prefer an oblique rotation method like promax over an orthogonal rotation method like varimax.

A: based on the advice of the reviewer we have explained in the paragraph our choice to use an orthogonal rotation method like varimax.

  1. 241: When reporting a reliability measure of the whole scale, it might be more viable to use stratified alpha or omega total based on a bifactor model as reliability measures.

A: Based on the reviewer's advice, we highlighted the result of Cronbach's Alpha in the paragraph.

  1. 247: The application of the Kolmogorov-Smirnov test seems questionable because it is unclear which procedure a normal distribution would be required.

A: Based on the advice of the reviewer we clarified the application of the Kolmogorov-Smirnov test, providing the reader with a clearer interpretation of the content of the study.

Minor comments:
16.    Please provide references to used software packages (117, 169).
17.    184: Remove “(n=317)” from the title and include it in the text.
18.    194: write “What is more”.
19.    237: write “Cronbach’s alpha”
20.    270: write “Burns and Grove”
21.    285: write “were not greater”

A: Based on the advice of the reviewer (from 16 to 21) we have modified all these things in the text.

Reviewer 3 Report

This manuscript is well-presented and offers elements a reader would note in development and validation of instruments. Please consider the following and incorporate where authors think are appropriate:

1. Please include results of normality tests (e.g., Shapiro-wilks) other than skewness and kurtosis) that were undertaken before EFA was performed.

2. Please include a paragraph describing whether there was any potential differential item functioning (for sub-groups of nurses described based on their clinical areas in lines 187-188). A simple t-test/anova could also be useful. This would indicate to readers the generalisability and usability of the instrument for various sub-groups of nurses. If there was no difference, that would be ideal as the instrument could be used across groups.

3. Please include a paragraph explaining the legitimate reasons as to why some error terms were allowed to be correlated in the CFA model. This is important, as the error terms should not be correlated for the sake of obtaining better fit indices.

Other than the above, the authors could consider extending this piece of meaningful work subsequently (e.g., to validate the data via modern test theory (e.g. Rasch) which is relatively more sample independent).

Author Response

REPLIES TO REVIEWER 3

This manuscript is well-presented and offers elements a reader would note in development and validation of instruments. Please consider the following and incorporate where authors think are appropriate:

  1. Please include results of normality tests (e.g., Shapiro-wilks) other than skewness and kurtosis) that were undertaken before EFA was performed.

A: Based on the advice of the reviewer we have put in the appendix a table with the results of the normality test.

  1. Please include a paragraph describing whether there was any potential differential item functioning (for sub-groups of nurses described based on their clinical areas in lines 187-188). A simple t-test/anova could also be useful. This would indicate to readers the generalisability and usability of the instrument for various sub-groups of nurses. If there was no difference, that would be ideal as the instrument could be used across groups.

A: As recommended by the reviewer we have inserted an anova table, in the text we have also put a paragraph describing the work accomplished.

  1. Please include a paragraph explaining the legitimate reasons as to why some error terms were allowed to be correlated in the CFA model. This is important, as the error terms should not be correlated for the sake of obtaining better fit indices.

A: based on the advice of the reviewer we have included a paragraph where we have highlighted why we justify the errors, improving the reading for the reader of the journal

Round 2

Reviewer 2 Report

The revised manuscript investigated the development and analysis of the nursing care and religious diversity scale (NCRDS) questionnaire. I have a few remaining comments:
1.    Table 3 still contains the “new item names”, while Table 4 and Figure 1 contain the “old item names”. Adapt it properly.
2.    241: It is inappropriate to say that varimax rotation is preferable when the loading structure is complex, while an oblique rotation like promax would opt for a simple loading structure. In contrast, both orthogonal and oblique rotation try to find a loading structure that is as simple as possible. However, oblique rotations allow the factors to be correlated, which is not the case for orthogonal rotations. I cannot imagine a situation where orthogonal rotations should be preferred for interpretation reasons. At least, the authors should not state incorrect statements in the manuscript. They can stick for whatever reason for the varimax method, however.
3.    306: write “Burns and Grove”. See also line 258 for a similar mistake.
4.    467: There is a typo when the reference starts.

Author Response

The revised manuscript investigated the development and analysis of the nursing care and religious diversity scale (NCRDS) questionnaire. I have a few remaining comments:
1.    Table 3 still contains the “new item names”, while Table 4 and Figure 1 contain the “old item names”. Adapt it properly.

  • As recommended by the reviser we have modified and corrected Table 3
  1. 241: It is inappropriate to say that varimax rotation is preferable when the loading structure is complex, while an oblique rotation like promax would opt for a simple loading structure. In contrast, both orthogonal and oblique rotation try to find a loading structure that is as simple as possible. However, oblique rotations allow the factors to be correlated, which is not the case for orthogonal rotations. I cannot imagine a situation where orthogonal rotations should be preferred for interpretation reasons. At least, the authors should not state incorrect statements in the manuscript. They can stick for whatever reason for the varimax method, however.
  • As recommended by the reviewer we corrected and explained the reason for the choice Varimax
  1. 306: write “Burns and Grove”. See also line 258 for a similar mistake.
  • As recommended by the reviewer we have corrected mistakes.
  1. 467: There is a typo when the reference starts.
  • As recommended by the reviewer we corrected the typo.

Reviewer 3 Report

-

-

Author Response

as indicated by the reviewer, we have sent to an English translator and editing to improve understanding of the text.